# Carbazole- Versus Phenothiazine-Based Electron Donors for Organic Dye-Sensitized Solar Cells

**DOI:** 10.3390/molecules30112423

**Published:** 2025-05-31

**Authors:** Daria Slobodinyuk, Alexey Slobodinyuk

**Affiliations:** 1Institute of Technical Chemistry Ural Branch of the Russian Academy of Sciences, Ac. Korolev 3, 614130 Perm, Russia; slobodinyuk.aleksey.ktn@mail.ru; 2Department of Chemical Engineering, Perm National Research Polytechnic University, Komsomolsky Prospekt, 29, 614990 Perm, Russia

**Keywords:** dye-sensitized solar cells, carbazole, phenothiazine, optical properties, electrochemical properties

## Abstract

Recently, research and development in the field of dye-sensitized solar cells has been actively advanced, as the technology constitutes a potential alternative to silicon-based photovoltaic devices. Modification of the molecular structure of the dye can enhance the adsorption on the TiO_2_ surface, improve the light absorption capacity, suppress the charge recombination, increase the electron injection rate, and thereby improve the overall performance of the solar cell. Carbazole and phenothiazine are rigid heterocyclic compounds containing nitrogen as a heteroatom with large π-conjugated skeletons. Phenothiazine differs from carbazole by the presence of sulfur as an additional electron-rich heteroatom. The inclusion of this heteroatom in the structure of the compounds can indeed improve the electron-donating properties, affect the conjugation, and thus affect the optical, electronic, and electrochemical properties of the chromophores as a whole. The difference in planarity when comparing carbazole with phenothiazine can be useful from several points of view. The planar structure of carbazole increases the degree of conjugation and the electron transfer capacity, which can increase the photocurrent of the cell. The nonplanar structure of phenothiazine helps to prevent π-stacking aggregation. This review comprehensively summarizes the progress in the field of synthesis of organic dyes for solar cells with an emphasis on the comparative analysis of two electron-donating moieties, carbazole and phenothiazine. In addition, the review describes in detail the relationship between the structure of the compounds (dyes), their properties, and the performance of solar cells.

## 1. Introduction

In recent years, there has been growing scientific interest in the development of new third-generation photovoltaic technologies such as dye-sensitized solar cells (DSSCs), organic solar cells (OSCs), and perovskite solar cells (PSCs), which are being developed as alternatives to silicon solar cells. Numerous studies in this area are related to the fact that these devices can compete with traditional silicon-based solar cells with their lower cost, ease of fabrication, and ability to be processed on flexible substrates [1]. However, despite the above advantages, all the mentioned solar cell technologies are not without their drawbacks. To date, OSCs have had difficulties due to the proper ordering of charge domains; PSCs suffer from their stability and sensitivity to moisture; and DSSCs require photosensitive materials with a broad absorption spectrum to demonstrate higher efficiency. In general, DSSCs have shown the best performance under practical outdoor conditions [2]. Dye-sensitized solar cells exhibit significant power generation performance throughout the day, even under low-intensity light, regardless of the incident angle of the light [3]. The photophysical characteristics of DSSCs are determined by the molecular structure of the dye. Therefore, they can be relatively easily tuned at the molecular level by varying the substituents and structural moieties. In addition, DSSCs are stable and efficient under harsh conditions such as low temperatures and low-light environments, even in the presence of moisture [4,5,6] Thus, DSSCs are still one of the most promising solar cell technologies, and research aimed at improving the efficiency of such solar cells is relevant today. Recent studies have shown that the best DSSCs based on metal-free organic dyes exhibited an energy conversion efficiency of 15.2% [7]. As is known, the efficiency of solar cells is determined by the molecular structure of the dye, which is a π-conjugated system with terminal electron-donating and electron-accepting fragments [8]. Many strategies have been considered in the literature to design highly efficient organic dyes depending on many factors such as the planarity and rigidity of donors and acceptors, bond type, conjugation length, and side chains [9,10,11,12,13]. Therefore, researchers face a significant challenge in designing electron donor moieties capable of highly efficient electron transfer and inhibition of molecular aggregation. In light of the previous considerations, phenothiazine (PTZ) and carbazole (CZ) are considered as effective options for donors (Figure 1) [14,15,16,17,18].

The introduction of carbazole and phenothiazine fragments into the structure of organic dyes is justified by their chemical stability and resistance to environmental influences. Substitution of hydrogen atoms at the nitrogen atom occurs easily, which makes it possible to improve solubility and adjust optical and electrical properties by introducing substituents. Carbazole and phenothiazine can be substituted at the third and sixth or seventh atoms to form 3,6- or 3,7-disubstituted derivatives, respectively, which are used to bind to other aromatic fragments either directly or through various π-conjugated bridges. The advantage of carbazole is that new synthetic methods now allow obtaining compounds in which carbazole fragments are included in the structure of the compound because of the participation of different carbon atoms and the nitrogen atom of the carbazole cycle: 2C-7C, 1C-8C, 9N-2C, 9N-3C, which changes the optical and electrochemical properties of the resulting compound over a wide range [19,20,21]. In addition, the structure of carbazole contains a diphenyl fragment, which makes the molecule flat and rigid. This structure promotes more efficient intramolecular charge transfer, which leads to high mobility of charge carriers. Phenothiazine, because of the presence of two electron-saturated heteroatoms (nitrogen and sulfur), has stronger electron-donor properties than carbazole. In addition, the introduction of this fragment into the structure of the compound prevents the formation of excimers (dimers in the excited state) that arise as a result of the interaction of excited and unexcited molecules and lead to a decrease in the fluorescence quantum yield [22]. This is explained, first of all, by the nonplanar structure of the phenothiazine molecule, which has a butterfly conformation in which the angle between the two benzene rings is 27.36 (9) [23]. The difference in planarity when comparing CZ with PTZ can be useful from different points of view. The planar structure of CZ increases the degree of conjugation and the ability to transfer electrons, which can increase the photocurrent [24]. The nonplanar PTZ block (butterfly-like skeletons) can prevent π-stacking aggregation [25,26]. Meanwhile, the addition of long alkyl chains such as 2-ethylhexyl group to the nitrogen atoms in the structures plays a role in creating steric hindrance and then inhibiting the aggregation of molecules [27].

The introduction of a carbazole or phenothiazine moiety has different effects on the optical, physical, physicochemical, and electrochemical properties of the compounds, which in turn determines the nature and efficiency of photoelectronic devices. Below is a review demonstrating a comparative analysis of compounds containing electron-donating carbazole and phenothiazine moieties in their structure, used as materials for dye-sensitized solar cells.

In order to determine the relationship between the structure of dyes, their properties and the efficiency of the device, a description of the structure of DSSCs and their operating principle is presented below.

The cells consist of two electrodes and an iodine-containing electrolyte. One electrode usually consists of a mesoporous dye-saturated oxide semiconductor (TiO_2_/ZnO/NiO) deposited on a transparent conductive substrate. The other electrode is a conductive glass plate onto which a reduction catalyst is deposited—metallic platinum or graphite [28]. The advantages of using TiO_2_ for the production of solar cells, compared with other materials, are the chemical resistance, nontoxicity, and low cost of the oxide. It features significant photoactivity, as well as a pronounced dependence of electrical properties on the surface morphology and type of crystal lattice. The band gap for TiO_2_, which does not absorb visible light, is 3.2 eV [29]. Figure 2 shows the operating principle of the cell [30].

Light absorption occurs because of a monolayer of dye chemically adsorbed on the surface of a semiconductor (e.g., TiO_2_) (1). After excitation by a light photon, the dye gives up an electron to the semiconductor (TiO_2_) (2), i.e., it passes into the conduction band of TiO_2_. The transition occurs very quickly and takes 10^−15^ s. In the semiconductor, the electron diffuses through the TiO_2_ film and reaches the glass electrode. The dye molecule is oxidized with the loss of an electron. The dye molecule is restored to its original state by obtaining an electron from the oxidation–reduction medium (electrolyte) (3). To complete the cycle, the electrolytic medium is neutralized by photoelectrons that reach the counter electrode through the external circuit (4). According to this principle, a dye-sensitized solar cell converts solar energy into electric current flowing through an external conductor [30,31]. To prevent liquid from leaking out during operation of the device, the cell is made hermetically sealed [31,32].

An analysis of the literature covering research in the field of dye-sensitized solar cells [28,33,34,35] allowed us to formulate a number of basic requirements for organic dyes, the fulfillment of which is necessary to achieve high efficiency of DSSCs.

It is believed that the sensitizer for DSSCs should have a high molar absorption coefficient (ε), which characterizes how strongly a substance absorbs light at a given wavelength.To ensure a high electron transfer rate, efficient conjugation of the donor and acceptor groups is necessary, and it is also important that the LUMO energy of the dye molecule is higher than the LUMO energy of TiO_2_. This will allow the HOMO electrons of the dye molecule to move to the LUMO orbital of the semiconductor (TiO_2_) rather than the dye when absorbing light quanta. The LUMO energy level is defined as the difference between the oxidation potential (E_ox_) and the band gap (E_g_^opt^), while it is worth noting that for TiO_2_ this value is −0.5 V vs. NHE [36].The photosensitizer (dye) in the oxidized state should be easily reduced by the electrolyte used (pair I_3_^−^/I^−^), i.e., the oxidation potential of the dye should be greater than the oxidation potential of the electrolyte (E_ox_ (I_3_^−^/I^−^) = 0.4 V).It is better for the dye to be covalently bound to the semiconductor surface; thus, the presence of “anchor” acid groups in its molecule is necessary.One of the factors causing a decrease in the energy conversion efficiency of many organic dyes in DSSCs is the formation of dye aggregates on the semiconductor surface. This affects light absorption via the internal filter effect. Therefore, to obtain optimal solar cell performance, it is necessary to avoid aggregation of organic dyes by structurally modifying the photosensitizer.Another important aspect when using organic dyes is their stability, which is usually lower than that of metal complexes. This is due to the formation of an excited triplet state and the formation of unstable radicals. The dye must be resistant to side photochemical and electrochemical processes and have noticeable thermal stability (withstand about 108 cycles of device operation).

The efficiency of using the approach described above is demonstrated below using examples of compounds containing carbazole and phenothiazine fragments that are structural analogues of each other.

## 2. D-π-A Type Organic Dyes

The D-π-A configuration is the most popular structure for organic dyes in DSSCs that facilitates charge separation and transfer. For example, compounds **1**, **2** (Figure 1) contain alkyl chains in their composition in order to increase the solubility of the compounds, as well as to prevent dye aggregation. The electron-withdrawing fragment of cyanoacrylic acid acts as an “anchor” group [37,38]. Compounds **1** and **2** differ from each other in the nature of the electron-donating fragment. The synthesis of D-A dyes **1**, **2** is a sequential implementation of three reactions—N-alkylation, formylation, and the Knoevenagel reaction (Figure 1).

The photophysical properties of compounds **1** and **2** are presented in Table 1. Compound **2**, containing a phenothiazine fragment, absorbs in a longer-wavelength region of the spectrum than the carbazole-containing compound **1**, which, in turn, leads to a decrease in the value of the band gap (**1**: E_g_^opt^ = 2.88 eV, **2**: E_g_^opt^ = 2.37 eV). It was found that the wavelength of the absorption maximum for dyes **1** and **2** adsorbed on a titanium dioxide surface underwent either a bathochromic shift (**1**: 33 nm) or a hypsochromic shift (**2**: 27 nm) compared with the absorption peaks for compound solutions, which in turn could lead to the formation of aggregates on TiO_2_.

Based on the results of electrochemical oxidation of compounds by cyclic voltammetry, it was proven that such structures had a more positive HOMO level than the electrolyte (I_3_^−^/I^−^). A quantitative estimate of this parameter is the oxidation potential (**1**: E_ox_ = 1.44 V, **2**: E_ox_ = 1.10 V, I_3_^−^/I^−^: E_ox_ = 0.4 V). This fact indicates that the oxidized compounds were easily reduced by the redox couple of the electrolyte (I_3_^−^/I^−^), which indicated the effective regeneration of dyes. To determine the LUMO energy levels, cyclic voltammetry data and absorption spectra data were used, which allowed calculating the excited state oxidation potential (E_ox_ − E_g_^opt^). The LUMO levels of compounds **1** and **2** showed more negative values than TiO_2_ (**1**: E_ox_ − E_g_^opt^ = −1.44 V, **2**: E_ox_ − E_g_^opt^ = −1.27 V, TiO_2_: E_ox_ − E_g_^opt^ = −0.5 B), which contributed to the effective electron injection process.

Because of the narrow absorption range of the carbazole-containing dye, the dye-sensitized solar cell manufactured on its basis demonstrated low power conversion efficiency (**1**: η = 0.42%). At the same time, despite the short conjugation chain in dye **2**, the presence of a phenothiazine fragment instead of a carbazole fragment contributed to a significant increase in this parameter (**2**: η = 5.5%).

The introduction of a 4-*tert*-butylphenyl fragment into the dye structure instead of alkyl chains is also one of the ways to provide steric hindrances in order to suppress intermolecular aggregation. Such an approach is presented in [39]. Dyes **11** and **12** contain in their structures 9-(4-*tert*-butylphenyl)-9*H*-carbazole or 10-(4-*tert*-butylphenyl)-10*H*-phenothiazine fragments connected via a thiophene–phenylene spacer to an electron-withdrawing fragment of cyanoacrylic acid (Figure 2). A multistep approach was used to synthesize such compounds, including initial *N*-arylation of the corresponding secondary amine (carbazole or phenothiazine) in the presence of copper salts (Ullmann reaction) or catalyzed by palladium complexes (Buchwald–Hartwig reaction) (Figure 2).

Further bromination of 9-(4-*tert*-butylphenyl)-9*H*-carbazole **3** or 10-(4-*tert*-butylphenyl)-10*H*-phenothiazine **4** afforded products **5** and **6**; subsequent chain extension was achieved by their initial reaction with 2-(tributylstannyl)thiophene under Stille reaction conditions to form 2-arylthiophenes **7** and **8** (Figure 2). Afterwards, reaction with BuLi afforded the corresponding lithium derivatives, the reaction of which with tri(n-butyl)tin chloride afforded the corresponding organotin compounds. Their cross-coupling with 4-bromobenzaldehyde introduced an additional phenylene spacer into compounds **7** and **8**. The final stage was the condensation of carbaldehydes **9** and **10** with cyanoacetic acid in the presence of catalytic amounts of ammonium acetate.

A study of the optical properties of compounds **11** and **12** revealed the presence of two absorption maxima corresponding to electron transitions in the electron-donor fragment (short-wave peak) and transitions characterizing intramolecular charge transfer (long-wave peak). It is worth noting that when replacing the 9-(4-*tert*-butylphenyl)-9*H*-carbazole fragment in compound **11** with 10-(4-*tert*-butylphenyl)-10*H*-phenothiazine, as in the previous example, a bathochromic shift of the wavelengths of the absorption maxima of the solutions of the compounds occurred, and the value of the molar extinction coefficient decreased (Table 2).

It is interesting that the wavelength of the absorption maximum of the CZ-containing compound **11** adsorbed on the titanium dioxide surface was slightly shifted to the short-wave region of the spectrum (10 nm) compared with the absorption peak for the compound solution. This is explained, first of all, by the fact that when the carboxyl group is bound to TiO_2_, its electron-acceptor properties decrease. In the case of compound **12**, containing a phenothiazine fragment in its structure, this difference reached 38 nm, which could lead to dye aggregation. In order to determine the efficiency of the solar cell, the HOMO and LUMO energies of dyes **11** and **12** were calculated. The results are presented in Table 2. The compounds demonstrated a sufficiently high LUMO level compared with TiO_2_ and a sufficiently low HOMO level compared with the ion pair of the electrolyte (I_3_^−^/I^−^). Thus, such structures provide efficient dye regeneration as well as an efficient electron injection process during the conversion of solar energy into electrical energy. Because compound **11**, containing a 9-(4-*tert*-butylphenyl)-9*H*-carbazole fragment in its structure, had the lowest HOMO level, a device made on its basis, according to the authors of [39], would demonstrate the most efficient charge regeneration.

Comparative analysis of the main characteristics of DSSCs showed that a device based on carbazole-containing dye were characterized by higher short-circuit current, higher open-circuit voltage, and higher field factor (**11**: *J*_sc_ = 14.63 mA cm^2^, V_oc_ = 0.685 V, FF = 0.67) than a device based on a phenothiazine analogue (**12**: *J*_sc_ = 14.12 mA cm^2^, V_oc_ = 0.68 V, FF = 0.64). As a result, the overall conversion efficiency of the former was better than that of the latter (6.70% versus 6.32%). The obtained values were quite close to the value of the well-known ruthenium complex N-719 (6.56%).

It was found that the use of condensed heterocyclic systems as a π-spacer has a number of advantages over their linearly linked analogs. Namely, such structures enhance light absorption and lead to the suppression of charge recombination [40]. This explains the interest in the synthesis of dyes **16** and **17**, containing 4,4-didodecyl-4*H*-cyclopenta [2,1-b:3,4-b’]dithiophene as a spacer (Figure 3). The starting carbaldehyde **13** was obtained from 4*H*-cyclopenta [2,1-b:3,4-b’]dithiophene by sequentially carrying out alkylation, formylation, and bromination reactions.

To introduce carbazole or phenothiazine fragments into the structure of the target product, the Suzuki reaction was used, the products of which (**14**, **15**) upon interaction with cyanoacetic acid led to the production of dyes **16** and **17**. Table 3 presents the results of studies of the optical and electrochemical properties of compounds **16** and **17**. The presence of a condensed heteroaromatic system in the structure of compounds **16** and **17** led to the leveling of the electron-donor properties of carbazole and phenothiazine, since both dyes exhibit absorption at the same wavelength, while, as in the case of compounds **1**, **2**, **11**, and **12**, compound **16**, containing a carbazole fragment, had the highest value of the molar extinction coefficient.

Experimentally calculated values of HOMO and LUMO energies proved the efficiency of using such dyes in DSSCs (**16**: η = 7.5%, **17**: η = 7.0%). Devices based on dyes **16** and **17** showed comparable PCEs with similar *J*_sc_ and V_oc_ values (**16**: *J*_sc_ = 15.2 mA cm^2^, V_oc_ = 0.691 V, **17**: *J*_sc_ = 12.9 mA cm^2^, V_oc_ = 0.774 V). This proves that for this type of structure, there is no major influence of the donor group on the PCE values of the devices.

## 3. D-A-π-A-Type Organic Dyes

In D-A-π-A-type compounds, an additional electron-withdrawing fragment allows for a significant facilitation of charge transfer from the donor to the acceptor located on the periphery of the molecule, as well as helping to tune the absorption region and HOMO and LUMO levels. In addition, compounds of this type have high photostability [41]. Figure 4 shows the synthesis of compounds **18** and **19**, the carbazole or phenothiazine fragments of which were linked to a cyanoacrylic acid fragment via an additional benzothiadiazole bridge [42,43]. It is worth noting that such structures can be obtained by two different methods. The first method involves carrying out four reactions—the Stille reaction, formylation, the Suzuki reaction, and the Knoevenagel reaction (Method A). The second method reduces the number of steps to three and involves a sequence of two Suzuki reactions, the final product of which, when reacted with cyanoacetic acid in chloroform with a catalytic amount of piperidine, leads to the formation of compounds **18** and **19** (Method B).

To clarify the question of how the introduction of an electron-withdrawing fragment as a π-spacer into the structure of dyes affects their photophysical properties, it is interesting to compare the properties of compound 18 with the characteristics of its analogue, compound **20** [41] (Figure 3).

The replacement of the thiophene ring in compound **20** with a benzothiadiazole ring (compound **18**) led to a slight shift in the wavelength of the absorption maximum to the long-wave region of the spectrum (**18**: λ_max_^abs^ = 507 nm, **20**: λ_max_^abs^ = 494 nm), which allowed expanding the spectral range of absorption. It is worth noting that the HOMO and LUMO energy values did not undergo significant changes (Table 4), but the experimentally obtained values of the power conversion efficiency differed by almost two times (**18**: η = 2.60%, **20**: η = 5.00%). It is known that the power conversion efficiency (η) is determined by the short-circuit current (*J*_sc_), open-circuit voltage (V_oc_), and fill factor (FF). For DSSCs based on dyes **18** and **20**, a significant difference was observed in the short-circuit current (**18**: *J*_sc_ = 5.78 mA cm^2^, **20**: *J*_sc_ = 10.35 mA cm^2^). The better *J*_sc_ of dye **20** can be attributed to its broader absorption profile and higher molar extinction coefficient. The introduction of a carbazole fragment into the dye structure instead of a phenothiazine fragment for this type of structure makes it possible to obtain a cell with a higher value of power conversion efficiency **19**: *J*_sc_ = 3.80 mA cm^2^.

Recently, dyes **21** and **22** were synthesized, containing carbazole or phenothiazine as an electron-donor fragment as well as benzothiadiazole as an auxiliary acceptor, benzene as a π-bridge, and cyanoacrylic acid as an anchor group [44] (Figure 5).

The synthesis of these dyes is shown in Figure 5 and included the initial preparation of 4,7-dibromobenzothiadiazole. The Buchwald–Hartwig reaction was used to introduce the phenothiazine fragment into the structure, and the Ullmann coupling reaction was used for the carbazole fragment. Further implementation of the palladium-catalyzed cross-coupling reaction of the obtained halides with 4-formylphenylboronic acid allowed us to obtain structures containing terminal aldehyde groups. The final stage of the synthesis of dyes **21** and **22** was the Knoevenagel reaction. The study of optical and electrochemical properties is presented in Table 5.

The synthesized dyes were characterized by a bathochromic shift of the absorption wavelength maxima in comparison with the compounds that did not contain a benzothiadiazole fragment [45]. It is worth noting that compound **22**, containing a phenothiazine fragment, had a longer-wavelength absorption region than the carbazole-containing compound **21**, but **22** was characterized by a lower value of the molar absorption coefficient. The HOMO level values for dyes **21** (1.10 V) and **22** (1.04 V) were higher than the corresponding value for the electrolyte (0.4 V for the redox pair I_3_^−^/I^−^), which ensured dye regeneration. At the same time, the HOMO level for the dye containing a phenothiazine fragment was less positive than that of the carbazole-containing dye, which was explained by the stronger electron-donor ability of the phenothiazine unit. On the other hand, the LUMO level values for dyes **21** (–1.39 V) and **22** (–1.16 V) were more negative than the conduction band of TiO_2_ (−0.5 V versus NHE), which ensured efficient electron injection into TiO_2_. The authors of the article tested the synthesized dyes in a DSSC. It was found that a cell based on a carbazole-containing dye was characterized by a higher value of energy conversion efficiency.

The DSSC based on carbazole dye **21** showed the highest PCE of 3.51% compared with the cell based on phenothiazine dye **22** (η = 1.76%). This was due to the relatively higher *J*_sc_ and V_oc_ values for the cell based on dye **21** (**21**: *J*_sc_ = 7.86 mA cm^2^, V_oc_ = 0.687 V; **22**: *J*_sc_ = 4.20 mA cm^2^, V_oc_ = 0.666 V). The better *J*_sc_ of compound **21** could be attributed to its broader absorption profile and higher molar extinction coefficient. Compared with dye **22**, the higher V_oc_ of dye **21** may have been due to the lower HOMO energy level, indicating a higher driving force for dye regeneration. Thus, more electrons could be accumulated in TiO_2_, indicating better suppression of charge recombination between TiO_2_ and electrolyte.

Compounds **23** and **24** contained dithieno [3′,2′:3,4;2»,3»:5,6]benzo [1,2-c]furazan as a π-spacer [46]. The advantage of introducing such an electron-withdrawing fragment is its rigid planar structure, which leads to efficient intramolecular charge transfer from the donor to the acceptor and allows reducing the reorganization energy of the dye molecule during photoexcitation. The synthetic approach to obtaining the compounds is presented in Figure 6. It uses a combination of the Stille reaction, formylation, bromination, and, in the final stage, condensation.

As in the case of dyes **16** and **17** containing a fused heterocyclic spacer, compounds **23** and **24** demonstrated absorption in the same region, but their distinctive feature was the molar extinction coefficient, which reached a higher value in the case of compounds of the D-A-π-A type (**23**, **24**) in the presence of a phenothiazine fragment in the structure (Table 6). The results of electrochemical oxidation for dyes **23** and **24** confirmed the pattern characteristic of compounds **1**, **2**, **11**, **12**, and **16–19**, namely, structures containing a carbazole fragment have a higher oxidation potential value and a deeper HOMO and LUMO level than FTZ-containing compounds, which is directly related to the power conversion efficiency (**23**: η = 1.42%, **24**: η = 5.98%). In addition, the low performance of the dye **23** cell was likely due to the fact that dye **23** is less effective in blocking electrolytes from approaching the TiO_2_ surface, resulting in a lower open circuit voltage (V_oc_). The dye **23** cell also had slower electron injection.

Recently, D-D-π-A-π-A quinoxaline dyes **25** and **26** were synthesized, differing from each other in the type of electron-donor fragment (Figure 4) [47].

Quinoxaline is a coplanar and rigid ring with strong electron-withdrawing ability due to two symmetric unsaturated nitrogen atoms in the pyrazine ring. In addition, the presence of the imine moiety increases the π-conjugation in the dye structure. It has been shown that the introduction of 2,3-diphenylquinoxaline into the dye structure reduces charge recombination by inhibiting intermolecular aggregation due to two separate phenyl rings attached to the quinoxaline block [48]. The multistep synthesis of dyes **25** and **26** is shown in Figure 7.

The results of a study on the optical and electrochemical properties of the dyes are presented in Table 7. The absorption spectra of the dyes demonstrated absorption bands in the region of 240–440 nm, which corresponded to the π-π*-electron transition due to the presence of aromatic fragments in the structure. Also in the low-energy region, peaks were observed at 440–660 nm, characterizing the intramolecular charge transfer between donor and acceptor fragments. The dyes were characterized by high values of the molar extinction coefficient (Table 7). It is important that these values were greater than those of the standard ruthenium dyes N719 and N3 (14,000 and 13,900 M^− 1^ cm^−1^, respectively). This fact proves that the presented dyes collected light significantly better than typical organometallic dyes.

For the presented type of structures, dye **26**, including a carbazole fragment, demonstrated a bathochromic shift of the absorption band, characterizing intramolecular charge transfer, in comparison with the phenothiazine-containing dye. The authors believed that this was due to the increased planarity of the carbazole fragment compared with the phenothiazine, which has a nonplanar butterfly shape. Parallel alignment of two benzene rings in the carbazole fragment provided smooth conjugation and, therefore, higher absorption.

Study of the electrochemical properties of the synthesized dyes showed that they were characterized by lower HOMO energies compared with the redox electrolyte (I_3_^−^/I^−^) (Table 7). This fact indicates that oxidative regeneration of the dye is possible through I^−^ in the DSSC electrolyte. On the other hand, the calculated LUMO values for the dyes were higher than that for TiO_2_, indicating possible electron injection from excited dyes into TiO_2_. As a result, all the dyes had sufficient fundamentality for their use as sensitizers in DSSCs.

## 4. Organic Dyes with a Star-Shaped Structure

Several approaches to dye design that prevent intermolecular aggregation of dye molecules during solar cell operation have been previously presented, namely the introduction of long alkyl chains or a 4-*tert*-butylphenyl fragment, as well as the use of a fused heteroaromatic system as a π-spacer. Dyes with a star-shaped structure are also of interest from the point of view of charge recombination suppression. For example, compounds **27** and **28** contain a central triphenylamine core, on the periphery of which there are carbazole or phenothiazine fragments, as well as a cyanoacrylic acid fragment [49]. The Ullmann reaction was used to synthesize these structures, the starting compound for which was 4-[bis(4-iodophenyl)amino]benzaldehyde, obtained in two stages from triphenylamine (Figure 8).

Table 8 presents the results of the analysis of the optical properties of dyes **27** and **28**. Both compounds had one intense absorption band in the visible region of the spectrum. Moreover, for the phenothiazine-containing compound **28**, as in the case of compound **23**, a higher value of the molar absorption coefficient was characteristic. The absorption spectra for solutions of compounds **27** and **28** in acetonitrile and the absorption spectra of compounds **27** and **28** adsorbed on the surface of TiO_2_ did not show a significant shift in the absorption wavelength, which indicates the prevention of aggregation of dyes on the titanium dioxide film (Table 8).

The experimentally obtained values of HOMO energies of compounds **27** and **28** (Table 8) proved that the oxidized dyes formed as a result of the corresponding injection of electrons into the conduction band of TiO_2_ accepting electrons from the electrolyte (I_3_^−^/I^−^), which led to their reduction. At the same time, the more negative value of the LUMO energy of the dyes (Table 8) compared with the LUMO energy of TiO_2_ (−0.5 V vs. NHE) indicates that the process of electron injection from the dye molecule in the excited state into the conduction band of titanium dioxide was energetically allowed. It was found that dyes **27** and **28** did not exhibit significant differences in photophysical properties. However, the FTZ-containing compound **28** was characterized by a higher value of power conversion efficiency (**27**: η = 3.26%, **28**: η = 4.54%), which was most likely due to its higher absorption capacity.

Study of the incident monochromatic photon-to-current conversion efficiency (IPCE) showed that the solar cells based on dyes **27** and **28** in the range of 400–500 nm showed high IPCE above 44%. Compared with dye **27**, the cell based on phenothiazine dye **28** gave a higher IPCE value, which implies that the dye would show a relatively large photocurrent in DSSCs. This fact may be related to the higher molar extinction coefficient for dye **28**. On the other hand, the IPCE values of **27** and **28** obviously decreased above 500 nm in the long-wavelength region, which could be attributed to the decrease in light collection for the dyes. These results are consistent with the absorption spectra in solution and on TiO_2_ films of the dyes.

Along with triphenylamine, the introduction of which into the structure of the molecule allows expanding the absorption region and reducing the tendency of aggregation on the TiO_2_ surface, triphenylethylene is of interest. For example, compounds **39–42** have a star-shaped structure, the central core of which is triphenylethylene connected to terminal carbazole or phenothiazine fragments as well as to the electron-withdrawing fragment of cyanoacrylic acid through electron-donating N-substituted carbazole or phenothiazine bridges (Figure 9) [50]. The starting compound for the synthesis of these structures was bis(4-fluorophenyl)ketone, which interacted with carbazole or phenothiazine; the products then entered into the Wadsworth–Emmons reaction, which led to the formation of the central triphenylethylene core (**31**, **32**) (Figure 9). The presence of the bromine atom in compounds **31** and **32** made it possible to obtain the corresponding boric acids **33** and **34** from them, from which carbaldehydes **35–38** were then synthesized under Suzuki reaction conditions. The final stage was the Knoevenagel reaction, which led to the target dyes **39–42**.

The spectra of dyes **39** and **41** had several absorption peaks in the ultraviolet region (**39**: λ_max_^abs^ = 238, 293, 341 nm, **41**: λ_max_^abs^ = 238, 293, 329, 341 nm), as well as one peak in the visible light region (**39**: λ_max_^abs^ = 412 nm, **41**: λ_max_^abs^ = 462 nm), and the long-wave absorption maximum of compound **41** was shifted to the red region of the spectrum (50 nm) (Table 9). The absorption spectra of dyes **40** and **42**, containing phenothiazine fragments at the periphery of the molecule, in dichloromethane contained three absorption peaks (**40**: λ_max_^abs^ = 258, 326, 466 nm, **42**: λ_max_^abs^ = 258, 335, 412 nm). It is also worth noting that for compound **40**, a bathochromic shift of the absorption maximum wavelength was observed compared with compound **42**, which was associated with the presence in its structure of a stronger electron-donating phenothiazine fragment, which was part of the π-spacer.

In the absorption spectra of dyes adsorbed on the surface of titanium dioxide, the absorption peaks were significantly shifted to the red region of the spectrum compared with the absorption wavelengths for solutions of compounds. Dyes **39–42** had a fairly deep HOMO level, about −5.0 eV, compared with the I^3−^/I^−^ ion pair (E_HOMO_ = −4.60 eV) [51], which indicated effective regeneration of the dye. It was found that the FTZ-containing compounds **40** and **41** had more negative values for their LUMO levels, which may have indicated their higher efficiency in terms of using dyes in solar cells, as evidenced by the experimentally obtained values of power conversion efficiency (**39**: η = 2.14%, **40**: η = 6.55%, **41**: η = 5.51%, **42**: η = 2.69%). In this case, the replacement of 9-hexyl-9*H*-carbazole in dye **39** with nonplanar 10-hexyl-10*H*-phenothiazine to construct the structure of dye **41** resulted in an improvement in the DSSC performance. In this case, further modification of the structure of compound **41**, namely the introduction of a more twisted triphenylethylenephenothiazine block instead of the triphenylethylenecarbazole one, allowed achieving a further increase in the DSSC performance based on dye **40**. Comparative analysis of dyes **39–42** showed that the more electron-rich and twisted nonplanar phenothiazine could be considered more favorable than planar carbazole in terms of its use in dye-sensitized solar cells. The nonplanar structure contributes to the slowing down of charge recombination and a reduction in dye aggregation, which leads to a higher V_oc_.

## 5. Organic Dyes of the A-D-A Type

The use of D-π-A configuration in organic dyes has some disadvantages; namely, the short length of π-conjugation contributes to a narrow absorption band. Although an increase in the length of π-conjugation expands the absorption region, this structure is not very stable when irradiated with high-energy photons. In addition, such a structure can cause aggregation and charge recombination, which can lead to a decrease in photoelectric characteristics.

Therefore, new configurations of organic dyes are currently being developed, such as A-D-A-type structures. Recently, Murali et al. synthesized two organic sensitizers with the A-D-A-D-A configuration, in which the donor was carbazole or phenothiazine, the auxiliary acceptor was benzothiadiazole, and the terminal group was cyanoacrylic acid (Figure 5) [52]. The presence of two terminal electron-acceptor fragments in the dye structure facilitated efficient electron injection.

The synthesis of the dyes consisted of the following reactions: N-alkylation, the Miyaura borylation reaction, the Suzuki cross-coupling reaction, the Vilsmeier–Haack reaction, and the Knoevenagel reaction (Figure 10).

The optical and electrochemical properties of organic dyes **43** and **44** are presented in Table 10. Both organic dyes exhibited two main absorption bands in the range of 340–410 nm and 420–540 nm. It is known that a wider absorption range contributes to better DSSC performance, since it allows collecting more photons. The observed red shift in the absorption maxima of dye **44** compared with those of dye **43** may have been due to the stronger electron-donating ability of the phenothiazine unit.

It is worth noting that for both organic dyes, the molar extinction coefficient values for the absorption maxima in the range of 420–540 nm, which characterize intramolecular charge transfer, were higher than those of the ruthenium dye N719 [53]. It has been proven that high molar absorption coefficients of organic dyes allow obtaining a thinner nanocrystalline film on their basis, which leads to better device performance. In addition, they promote the diffusion of electrolyte in the film and reduce the possibility of recombination of light-induced charges during transport [54,55].

In the absorption spectra of the dyes adsorbed on the transparent TiO_2_ thin film (2.5 mm), the absorption maxima of dyes **43** and **44** showed red shifts of about 18 and 12 nm, respectively, compared with their dissolved state. The observed red shifts of the absorption maxima and broadening of the absorption spectra for both dyes were possibly due to the formation of J-aggregates on the thin films.

To evaluate the possibility of electron transfer from the excited organic dye molecule to the conduction band of TiO_2_ and from the redox couple in the electrolyte to the oxidized dye molecule, cyclic voltammetry measurements were performed for both dyes. The results are presented in Table 10.

It was found that for organic dyes **43** and **44**, the ground state oxidation potential (E_ox_) was observed at 0.91 and 0.62 V, respectively, which was significantly higher than the iodide/triiodide oxidation–reduction potential (0.4 V). In addition, it is worth noting that the phenothiazine moiety had a significant effect on the oxidation potential of the dye, which led to a lower E_ox_ value for 44 compared with **43**. The calculated excited state oxidation potentials for the dyes were more negative (**43**: E_ox_* = 1.45 V, **44**: E_ox_* = 1.56 V) than the conduction band edge energy level of TiO_2_ (0.5 V). This fact indicates that electrons from the excited dye molecules can be effectively injected into the conduction band of TiO_2_. In general, the energy levels of dyes **43** and **44** were in good agreement with the requirements for efficient electron transfer in DSSC.

Solar cells prepared based on dyes **43** and **44** exhibited power conversion efficiencies of 4.35% and 6.87%, respectively. Moreover, the PCE values of the cells did not even show significant changes after aging at room temperature for 7 days. The presence of a phenothiazine moiety in compound **44** resulted in an improved *J*_sc_ value compared with compound **43** (**43**: *J*_sc_ = 7.51 mA cm^2^, **44**: *J*_sc_ = 13.1 mA cm^2^). The observed high *J*_sc_ value of the dye-**44**-based cell can be attributed to its stronger ability to collect power in the longer wavelength region. On the other hand, the dye-**44**-based device exhibited a comparatively lower V_oc_ value than dye **43** due to its higher oxidation potential value (**43**: V_oc_ = 0.762 V, **44**: V_oc_ = 0.728 V). Overall, it is evident that the high short circuit density value of **44** resulted in a higher efficiency of 6.87%. Furthermore, the observed high performance of the phenothiazine dye-based device may have been due to its lower tendency to aggregate.

It is worth noting that rhodanine-3-acetic acid can be used as an “anchor” electron-withdrawing group in dyes instead of cyanoacrylic acid. For example, compounds **45** and **46** (Figure 11), which are A-D-A-type structures, contain terminal rhodanine-3-acetic acid fragments linked to an N-ethylcarbazole or N-ethylphenathiazine fragment via a methylene bridge [56].

An alternative approach to the Vilsmeier–Haack reaction, in which the starting dicarbaldehydes were obtained in two stages, was used to synthesize these structures. Initially, the reaction product of imidazole with trifluoroacetic anhydride interacted with N-substituted carbazole or phenothiazine to form intermediate compounds containing trifluoroacetyl groups, the further hydrolysis of which led to the production of dicarbaldehydes. The final stage of the synthesis of compounds **45** and **46** was the Knoevenagel reaction.

A comparative analysis of the photophysical characteristics of compounds **45** and **46** is presented in Table 11. As expected, the presence of a phenothiazine fragment in the structure of compound **46** shifted the absorption band to the long-wavelength region of the spectrum compared with the CZ-containing compound **45**, which, in turn, led to a decrease in the value of the band gap (**45**: E_g_^opt^ = 2.56 eV, **46**: E_g_^opt^ = 2.18 eV).

Compound **45** was characterized by a more positive HOMO level, which indicated effective dye regeneration, but compound **46** had an effective electron injection process due to a deeper LUMO level. A study of the volt–ampere characteristics of solar cells found that the FTZ-containing compound **46** had the highest power conversion efficiency (**45**: η = 2.81%, **46**: η = 4.91%). The improved efficiency exhibited by the dye-based sensitizer **46** was due to the stronger electron-donating ability of the phenothiazine block in transferring electrons from the phenothiazine block to the rhodanine-3-acetic acid group and the possible vectorized photon-induced charge transfer of the phenothiazine block toward the electrodes. According to the authors of the paper, this characteristic could not only suppress the interaction between molecules, leading to energetic quenching of excited states, but suppress I_3_^−^ ions in the electrolyte on the TiO_2_ surface, which favored a higher V_oc_.

## 6. Conclusions

Carbazole and phenothiazine are important structural fragments in the creation of materials for dye-sensitized solar cells. This is explained, first of all, by the fact that compounds containing CZ- or FTZ-fragments have high thermal and morphological stability. In addition, conjugated systems containing electron-donating carbazole and phenothiazine fragments in combination with the electron-accepting fragment of cyanoacrylic acid have a sufficiently deep HOMO level compared with the I_3_^−^/I^−^ ion pair and are characterized by a more negative LUMO energy value than that of TiO_2_. The first fact proves that oxidized dyes formed as a result of the corresponding electron injection into the conduction band of TiO_2_ accept electrons from the electrolyte (I_3_^−^/I^−^), which leads to their reduction. The second fact indicates that the process of electron injection from the dye molecule in an excited state into the conduction band of titanium dioxide is energetically allowed. As a result, the use of such structures as materials for dye-sensitized solar cells will allow obtaining devices with high power conversion efficiency.

## Data Availability

The data that support the findings of this study are available from the corresponding author under reasonable request.

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
