# Peer review of "Carbazole- Versus Phenothiazine-Based Electron Donors for Organic Dye-Sensitized Solar Cells"

_molecules, 2025, doi:10.3390/molecules30112423_

Round 1

Reviewer 1 Report

Comments and Suggestions for Authors

The article explores advancements in organic dye synthesis for dye-sensitized solar cells, focusing on the comparison between carbazole and phenothiazine as electron-donating moieties. Carbazole, with its planar structure, enhances electron transfer and photocurrent due to increased conjugation. In contrast, phenothiazine's non-planar structure prevents aggregation, thereby improving stability.

The study underscores the importance of these structural fragments in enhancing light absorption, reducing charge recombination, and improving overall solar cell performance. This comprehensive review emphasizes the relationship between dye structure, properties, and solar cell efficiency, highlighting their potential for future photovoltaic applications.

The manuscript presents a clear and well-structured review on the role of carbazole and phenothiazine derivatives in the design of organic DSSCs. The authors provide a comprehensive and up-to-date overview of the influence of molecular structure on the optical, electronic, and electrochemical properties of dyes, and effectively correlate these features with the photovoltaic performance of DSSCs. I believe that the manuscript deserves publication in Molecules after minor revisions. I kindly request the authors to consider the following suggestions:

Comments:
1) Although the manuscript is generally well-written, a careful proofreading is recommended to correct any grammatical issues.

2) Please check the validity of reaction (4): it should be corrected as I3​+2ecb→ 3I

3) Please correct the following sentence in the introduction: "The transition occurs very quickly and takes 10-15 s. In the semiconductor, the electron diffuses through the TiO2 film and reaches the glass electrode." Do the authors mean 10-15 s?

4)  The authors should carefully review all the reaction schemes, as some contain incorrect or inconsistent font usage

5) The authors are encouraged to expand the literature to better highlight the role of carbazole and phenothiazine, including recent theoretical studies that clarify how molecular structure influences the electronic properties of donor–acceptor systems. In particular, J. Phys. Chem. C (2024, 128, 11998–12009) and Struct. Chem. (2024, 35, 1843–1863) offer detailed computational analyses of carbazole-based D–A–D architectures, demonstrating how features such as planarity and electronic decoupling govern intersystem crossing efficiency and excited-state dynamics.

Author Response

Dear Reviewer!

Thank you greatly for the attentive and careful work with our paper!

We have tried to do our best and to consider all the remarks You have made in the Report.

Corrections in the article based on comments are highlighted with a yellow marker.

Responses to your comments are provided in a separate file.

Reviewer 2 Report

Comments and Suggestions for Authors
  1. In the “introduction” part, it would be better to provide an illustration graph to describe the structure of a typical dye-sensitized solar cell, affiliated with a working mechanism.
  2. In Figure 2, the cell was named as “Graetzel electrochemical cell”. Although it has the same meaning as DSSCs, it is better to unifying the terminology of the cells.
  3. This review mainly discussed the synthesis and energy levels distribution of carbazole and phenothiazine in detail. However, the preparation and performance of DSSC based the two dyes were seldom inferred and discussed. For example, how the change of molecular structures influence/correlate with the performance parameters of DSSC should be analyzed. In my opinion, this part should also be equivalently important for the review.
  4. In some places, there only existed one sentence in a paragraph (line 41-42, 158-159, 209-212, 290-293, etc), and the arrangement seems a little wired.
  5. In Tables, the property items about the dyes are few. More properties index should be added. It is oversimplified.
  6. The latest reference cited in the manuscript was published in 2023 (one paper) and 2024 (one paper). The referred literatures are old.
  7. In a typical DSSC, the photoanode is the place where incident light passes through the cell for triggering absorption by the dyes. What is the role of a photocathode? Never heard the usage of a photocathode in a DSSC (in line 111).
  8. New solar energy utilization technology is emerging and flourishing, such as perovskite solar cells and organic photovoltaic devices. How about the prospects of DSSCs, and what aspects can scientific researchers do their effect for?

Author Response

Dear Reviewer,

Thank you very much for your detailed analysis of our manuscript and valuable comments!

We have tried our best to take into account all the comments you made in the report.

Responses to your comments are provided in a separate file.
